# Validation of Non-Invasive Preimplantation Genetic Screening Using a Routine IVF Laboratory Workflow

**DOI:** 10.3390/biomedicines10061386

**Published:** 2022-06-11

**Authors:** Ni-Chin Tsai, Yun-Chiao Chang, Yi-Ru Su, Yi-Chi Lin, Pei-Ling Weng, Yin-Hua Cheng, Yi-Ling Li, Kuo-Chung Lan

**Affiliations:** 1Graduate Institute of Clinical Medicine, College of Medicine, Kaohsiung Medical University, Kaohsiung 80708, Taiwan; ninytsai@gmail.com; 2Department of Obstetrics and Gynecology, Pingtung Christian Hospital, Pingtung 90053, Taiwan; 3Center for Menopause and Reproductive Medicine Research, Kaohsiung Chang Gung Memorial Hospital, Kaohsiung 83301, Taiwan; maurenlab15@gmail.com (Y.-C.C.); lingpay@gmail.com (P.-L.W.); justjudykimo@gmail.com (Y.-H.C.); 4Department of Obstetrics and Gynecology, Kaohsiung Chang Gung Memorial Hospital, Chang Gung University College of Medicine, Kaohsiung 83301, Taiwan; arena0920@gmail.com (Y.-R.S.); icsi@cgmh.org.tw (Y.-C.L.); pilotli9093@gmail.com (Y.-L.L.); 5Department of Obstetrics and Gynecology, Jen-Ai Hospital, Taichung 41257, Taiwan

**Keywords:** embryo selection, non-invasive preimplantation genetic screening, spent culture medium, cell-free DNA, time-lapse

## Abstract

Embryo selection is needed to optimize the chances of pregnancy in assisted reproduction technology. This study aimed to validate non-invasive preimplantation genetic testing for aneuploidy (niPGT-A) using a routine IVF laboratory workflow. Can niPGT-A combined with time-lapse morphokinetics provide a better embryo-selection strategy? A total of 118 spent culture mediums (SCMs) from 32 couples were collected. A total of 40 SCMs and 40 corresponding trophectoderm (TE) biopsy samples (*n* = 29) or arrested embryos (*n* = 11) were assessed for concordance. All embryos were cultured to the blastocyst stage (day 5 or 6) in a single-embryo culture time-lapse incubator. The modified multiple annealing and looping-based amplification cycle (MALBAC) single-cell whole genome amplification method was used to amplify cell-free DNA (cfDNA) from the SCM, which was then sequenced on the Illumina MiSeq system. The majority of insemination methods were conventional IVF. Low cfDNA concentrations were noted in this study. The amplification niPGT-A and conventional PGT-A was 67.7%. Based on this study, performing niPGT-A without altering the daily laboratory procedures cannot provide a precise diagnosis. However, niPGT-A can be applied in clinical IVF, enabling the addition of blastocysts with a better prediction of euploidy for transfer.

## 1. Introduction

Embryo selection is needed to optimize the chances of pregnancy and simultaneously reduce the incidence of multiple pregnancies and their accompanying problems [1]. Preimplantation genetic testing-aneuploidy (PGT-A) is applied in clinical screening for family genetic disorders or in cases of advanced maternal age to lower the rate of miscarriages and shorten the time intervals to pregnancy [2,3]. However, current limitations of PGT-A still exist owing to the invasive nature of biopsy procedures, lack of long-term bio-safety evaluation, and technical expertise requirements [4]. Another concern of embryo biopsies is “embryo mosaicism.” As only a small portion of cells can be collected from the embryo, this limits the genetic diagnosis of the embryo’s DNA integrity [5].

Developmental plasticity can be shaped by DNA, epigenetic modifications, and embryonic stimulation [6]. Artificial manipulation of PGT-A (e.g., intracytoplasmic sperm injection (ICSI), assisted hatching, trophectoderm (TE) biopsy) during early embryonic development may lead to adaptations in the anatomy, physiology, and metabolism of various organ systems, thereby affecting disease susceptibility or epigenetic modifications [6,7]. Therefore, non-invasive embryo selection has become more intriguing [8,9]. Current non-invasive embryo selection mainly relies on embryo morphology and morphokinetic assessment [10]. The benefit of blastocyst transfer has contributed to the natural selection of good-quality embryos and is reported to result in high implantation rates while lowering the multiple gestation rates following IVF [11]. Morphokinetic observation of time-lapse imaging is a non-invasive method of embryo selection [12,13]. Recently, different algorithms were developed to correlate embryo kinetics to blastocyst formation, implantation potential, embryo ploidy, and live birth rate [14,15]. However, the European Society of Human Reproduction and Embryology (ESHRE) time-lapse working group concluded that the relevant parameters were reported, the desired results have not been achieved, and more research is required [12]. Moreover, the predictive value of time-lapse analysis for embryo ploidy status remains difficult to determine [14,15].

In addition to the evolving advances in morphological evaluation, including morphokinetics, non-invasive methods based on the detection of molecular markers and cell-free DNA are present in the SCM of the embryo developed [16,17,18]. Reproductive scientists have dedicated more efforts to non-invasive preimplantation genetic testing for aneuploidies (niPGT-A) [19,20,21,22,23,24,25,26,27,28] to avoid embryo biopsies, thereby limiting the associated risks. Briefly, cell-free DNA (cfDNA) in spent culture medium (SCM) and/or blastocoele fluid (BF) [29] was collected and sequenced on a next-generation sequencing (NGS) platform, or analyzed by array comparative genomic hybridization (aCGH) following whole-genome amplification (WGA) in the mainstream pipeline [25]. According to some research groups, niPGT-A has comparable diagnostic efficiency to TE-biopsy PGT-A [22,24,27]. Moreover, niPGT-A using SCM may be more reliable than initial TE biopsy for predicting karyotypes of ICM in mosaic embryos [22,30,31].

To date, the handling process of niPGT-A has involved some modified workflow in the IVF lab [32] that may oppose the aim of a “non-invasive” study. For example, gamete fertilization was carried out mainly via ICSI (86.7%) [33] to prevent parental contamination. The culture medium was reduced to 10–20 μL to obtain a higher DNA concentration [19,22,23], embryos were cultured and extended to day 6/7 to obtain more embryonic cfDNA [20], and thawing vitrified embryos, and extended culture [22,25]. Furthermore, high rates of false-positive diagnoses of human embryos with normal pregnancy potential have led to the non-use and/or disposal of embryos. Until further validation studies are performed, niPGT-A should not be recommended for use in routine clinical settings as a diagnostic procedure [34,35].

We propose the use of niPGT-A as a reference for embryo selection priorities before transfer rather than a tool for precise diagnosis. In this study, we aimed to validate niPGT-A using our routine IVF laboratory workflow. Of note, our embryo-handling protocol or routine procedures were not optimized to fit the suggested modified workflow [22]. ICSI was not routinely used, and the amount of culture medium used for this assessment was not reduced. We characterized cfDNA and determined whether niPGT-A would change the priority of embryo selection under this setting. niPGT-A combined with time-lapse parameters may serve as a tool for prioritizing embryos.

## 2. Materials and Methods

### 2.1. Study Design

This is a prospective cohort study conducted at IVF KCGMH from March 2019 to July 2020. All IVF stimulation protocols and laboratory workflow were performed according to standard clinical practices as previously published [36,37]. All couples completed the standard infertility workup. Expected normal or hyper-responders that require embryo selection were included in this study after informed consent. Embryos obtained from these couples must be cultured in time-lapse. A total of 118 fresh SCMs were collected from the extended culture cycles of 32 couples during the study period.

### 2.2. Embryo Management and Embryo Culture

All embryos were inseminated with conventional IVF or ICSI and cultured to the blastocyst stage (day 5 or 6 post-insemination) in a single embryo culture time-lapse system (CCM-IVF, ASTEC, Fukuoka, Japan). Embryos were scored on days 1, 3, and 5/6. Blastocysts were graded according to the Gardner and Schoolcraft system [38], which is based on the degree of blastocyst expansion and morphological appearance of the inner cell mass and TE cells. Top-quality blastocysts were defined as 3AA, 4AA, 5AA, or 6AA [37].

### 2.3. Spent Culture Medium Collection

Embryos were cultured in G1^™^ medium (Vitrolife Sweden AB, Vastra Frolunda, Sweden) on days 1–3 and G2^™^ medium (Vitrolife Sweden AB) on days 3–5 or 6. SCM was collected on day 5/6. The incubation times in this study were two days for day 5 blastocysts, and three days for day 6 blastocysts. If the embryos were arrested during development, the SCM was collected on day 3. Embryos with the same number of blastomeres at two sequential times (24 h) were considered to be experiencing developmental arrest [39]. Different Pasteur pipettes were used for each embryo to prevent cross-contamination of media. Of note, our program routinely offers elective single blastocyst transfer to couples with three or more 8-cell embryos on day 3 [11]. Surplus embryos were routinely cultured to the day 5/6 (blastocyst stage) for freezing. An infrared laser (Hamilton Thorne Biosciences, Beverley, MA, USA) was used on an expanded blastocyst to laser a minor breach over the cellular junction of the TE to release the blastocoel fluid into the culture medium before vitrification (artificial shrinkage) [40]; this breach was far from the inner cell mass. After the embryo was removed, the released blastocoel fluid (only if artificial shrinkage is performed) mixed with the culture media was transferred to RNase-DNase-free PCR tubes. All blastocyst media (25–30 μL) were transferred from each embryo into RNase-DNase-free PCR tubes containing five μL of cell lysis buffer (Yikon Genomics, Shanghai, China) [27]. A blank medium control consisted of embryo culture medium without contact with the embryo. As a negative control, the same amount of blastocyst culture medium was collected but was not used for embryo culture. All collected samples were frozen immediately in liquid nitrogen and stored at −80 °C until use in the niPGT-A assay [27].

### 2.4. Whole-Genome Amplification and DNA Sequencing

TE biopsies and D5/D6 SCM from each embryo were transferred into RNase–DNase-free medium. The PCR tubes were stored at −80 °C until DNA amplification. Arrested whole embryos were used for chromosome ploidy analysis. Whole-genome amplification (WGA) and sequencing libraries were performed using the ChromInst Library Preparation Kit or NICSInst Library Preparation Kit, which is based on a modified multiple annealing and looping-based amplification cycles (MALBAC) method, according to the manufacturer’s protocol (Yikon Genomics, Shanghai, China) [27]. After the libraries constructions were performed, the produced DNA amplicons were subjected to TapeStation Bioanalyzer 4200 and Qubit dsDNA HS Assay kit (Life Technologies, Carlsbad, CA, USA) for measuring amplicon size and concentration, respectively. Only the DNA amplicon samples with the qualified size in length and concentration were applied for further assay (Quality control, QC pass). DNA library samples were pooled into batches of 24 samples. The libraries were sequenced using the Illumina MiSeq platform, which yielded approximately 2 million sequencing reads for each sample. We counted the read numbers along the whole genome, with a bin size of 1 Mb.

### 2.5. Genetic Data Analysis

ChromGo software (Yikon Genomics, Shanghai, China) was used to automatically analyze the sequencing data and report chromosomal abnormalities. This software allows the evaluation of entire chromosomes, assessment of the short and long arms of each chromosome, detection of deletions or duplication of >10Mb, and the detection of embryo sex and the presence of mosaicism [27].

### 2.6. Time-Lapse System

The time-lapse imaging system (TLIS) (CCM-iBis, ASTEC Co., Fukuoka, Japan) had a red LED as the light source (peak wavelength: 623 nm) and a CCD sensor camera unit connected to a 10× biological microscopes. The resolution of the camera was 1.3 megapixels, and the size of the observed area was 4.86 mm × 3.62 mm. TLIS was performed every 15 min, and images of each embryo were captured at 11 focal planes separated by five μm. If an embryo is cultured from day one (2PN) to day five, 5225 photos (approximately 14.3 G) were obtained. Kinetic variables included the time to 2 (t2), 3 (t3), 4 (t4), and 5 (t5) cells; the length of the second (cc2 = t3–t2) and third (cc3 = t5–t3) cell cycle; the synchrony in the division from 2 to 4 cells (s2 = t4–t3); the interval t5–t2; direct cleavage (DIR); and reverse cleavage [41,42,43,44]. The morphokinetic variables were retrospectively reviewed. Normal morphokinetic intervals were defined as follows: cc2 ≤ 11.9 h [41], cc3 11.7–18.2 h [44], (t5–t2) > 20.5 h [44]. The DIR was defined as (cc2 = t3–t2) < 5 h [45].

### 2.7. Embryo Transfer

The embryo selection was guided by morphology grade and/or the results of the TE biopsy. The niPGT-A results and time-lapse parameters were blinded to the primary care doctor when embryo transfer.

### 2.8. Statistical Analysis

Ploidy concordance was defined when euploid–euploid or aneuploid–aneuploid was obtained from the result of TE biopsy and SCM from the two sample types. The performance of niPGT-A, time-lapse parameters and morphology assessment was quantified in terms of the area under the receiver operating characteristic (ROC) curve (AUC), specificity, and sensitivity. The positive predictive value (PPV), negative predictive value (NPV), sensitivity, and specificity were calculated for euploidy versus aneuploidy [46]. Continuous data are presented as mean ± standard deviation (SD). Student’s *t*-test was used to compare continuous data. Categorical variables are reported as proportions and were compared using the Chi-squared or Fisher’s exact test, as appropriate. A binary variable was defined by the time interval value inside, and vice versa for outside the optimal range. Logistic regression analysis was employed to calculate the odds ratio (OR). The OR of the effect of all binary variables generated for embryonic chromosomal normality is expressed in terms of the 95% confidence interval (95% CI) and significance. All tests of significance were two-tailed, with *p* < 0.05 indicating statistical significance. All statistical analyses were performed using SPSS for Windows version 18.

## 3. Results

### 3.1. Characterization of Cell-Free DNA

A total of 118 SCMs were collected from 32 couples for cfDNA analysis. Maternal age ranged from 23 to 42 years (mean age 34.8 ± 4.9 years). The post-amplified mean DNA concentration was 1.97 ng/μL, the median concentration was 1.5 ng/μL (interquartile range [IQR] 1.0–2.5 ng/μL), and the failed QC rate was 10.2% (12/118). Sixty-four percent (*n* = 75) of SCM was derived from IVF while 36% (*n* = 43) was derived from ICSI. Maternal age was found to be comparable between the two groups (IVF, 34.5 y/o; ICSI, 35.4 y/o; *p* = 0.35). After WGA, the amount was not found to significantly differ between IVF (2.1 ng/μL) and ICSI (1.8 ng/μL). The failed QC was not significantly higher in the IVF group (Table 1). Global parental DNA contamination was observed in IVF (21.3%) and 18.6% in ICSI (*p* = 0.397). Maternal and paternal contamination was observed in the IVF group, whereas no paternal contamination was noted in the ICSI group.

After excluding 12 failed QC culture media, 106 SCMs (day 5 = 85, day 6 = 21) were analyzed. After WGA, the DNA amount was found to be higher on day 6 SCM (D6, 2.56 ± 1.7 ng/μL; D5, 1.86 ± 1.6 ng/μL; *p* = 0.05). Other parameters, including age, insemination methods, embryo quality, QC pass rate, and contamination rate, were comparable between the two groups.

### 3.2. Concordance of the Results from Nipgt-A and Their Corresponding Embryos/TE Biopsies

The flow chart outlining the steps used to evaluate the concordance is presented in Appendix A. A total of 40 SCMs and 40 corresponding TE biopsy samples (*n* = 29) or arrested whole embryos (*n* = 11) from seven couples were analyzed for concordance. The mean age of the couples was 38.4 years old (32, 38, 39, 39, 40, 40, 41 years). Based on their biopsy indications, there was one case of repeated IVF failure and six cases of advanced maternal age. The SCM of the respective embryos (i.e., niPGT-A samples (*n* = 40)) were collected for the analysis of cell-free DNA. All 40 samples were amplified after WGA and analyzed using NGS. The results were categorized as euploid or aneuploid. Detailed information for all samples is provided in Table 2. Overall, the aneuploidy rate from TE biopsy/whole embryo was 26/40 = 65%. The results revealed that 4/40 = 10% of TE PGT-A was mosaicism. Further, 73% (8/11) of arrested embryos were aneuploids in PGT-A. The cfDNA amount after MALBAC-WGA was higher in euploid samples (2.11 vs. 1.53 ng/μL, *p* = 0.05) than in aneuploid samples (Appendix A). Maternal age was found to be comparable between the two groups (euploidy, 38.3 y/o; aneuploidy, 39.4 y/o; *p* = 0.12). Based on subgroup analysis, the cfDNA amount after MALBAC-WGA was significantly lower in aneuploid samples than euploidy samples (1.43 vs. 2.09 ng/μL, *p* = 0.02) from females older than 35 years old (*n* = 37) (Appendix A).

ROC analysis resulted in an AUC of 0.73 ± 0.09 for the niPGT-A, 0.70 ± 0.09 for the time-lapse parameters (CC3 11.7–18.2 h), 0.59 ± 0.10 for morphology grading for the prediction of euploidy in PGT-A (Figure 1).

Nine samples from the SCMs were excluded owing to the low quality of the DNA in the NGS analysis. Accordingly, 31 samples were included in the concordance analysis (Appendix A). The overall ploidy concordance rate between niPGT-A and PGT-A was 21/31 (67.7%). The full concordance rate in the present study was 8/31 = 25.8%. The ploidy concordance rate was higher in ICSI samples (ICSI, 11/15 = 73.3%; IVF, 10/16 = 62.5%; *p* = 0.52). To evaluate the diagnostic accuracy of niPGT-A, we calculated the sensitivity (i.e., the proportion of aneuploid embryos that were correctly identified in SCM) and specificity (i.e., the proportion of euploid embryos that were correctly identified in SCM) of niPGT-A using the results of PGT-A as the gold standard. By comparing each SCM with their corresponding TE/whole embryos, the sensitivity and specificity of niPGT-A were 11/19 = 57.9% and 10/12 = 83.3%, respectively. The positive predictive value (PPV) and negative predictive value (NPV) of niPGT-A in identifying chromosomal abnormalities were 84.6% and 55.6%, respectively.

Excluding the arrested embryos, the overall ploidy concordance rate was 58.3%. Further, the PPV and NPV of niPGT-A in identifying chromosomal abnormalities were 62.5% and 56.2%, respectively.

### 3.3. Embryo Section Algorithm Using NiPGT-A and/or Time-Lapse Morphokinetics

Euploid embryos had a significantly larger normal % of CC3 than aneuploid embryos based on the time-lapse parameters (78.6% vs. 46.2%, *p* = 0.048) (Appendix A). The incidence of reverse cleavage in this study was 7.5% (3/40). All reverse conditions occurred in aneuploid embryos. In the full euploid concordance situation (*n* = 9), the corresponding time-lapse data had significant normal time-interval indicators (CC2, CC3, and T5–T2) and no reverse cleavage (Table 3). Of these parameters, CC3 was the most significant (*p* < 0.001).

niPGT-A was the only significant variable in the prediction of embryo aneuploidy (in PGT-A) in logistic regression analysis (odds ratio [OR] = 8.28; 95% confidence interval [CI], 1.73–39.65; *p* = 0.008). Based on these results, an algorithm for embryo selection (Figure 2) was proposed to classify embryos.

Thirty-one transfer cycles (mean maternal age: 35.0 ± 4.4 y/o) with a total of 47 embryos were transferred (47 with SCM euploidy, nine with SCM aneuploidy, and one with no signal), resulting in 22 pregnancies and five abortions. Among the 22 pregnancy cycles, 20 (90.9%) pregnancies had at least one SCM euploid embryo transfer while 15 (68.2%) had a live birth pregnancy with at least one SCM euploid embryo transfer.

By examining the niPGT-A and/or time-lapse parameters using the proposed embryo section algorithm, we found that 19.3% will change ET priority if niPGT-A was known in advance; 22.6% will change ET priority if combined niPGT-A and TLM parameters in advance (Appendix A).

## 4. Discussion

To our knowledge, this is the first study to explore cfDNA in SCM without altering the routine IVF laboratory workflow and validate truly non-invasive embryo selection using time-lapse morphokinetics and niPGT-A.

In the present study, IVF, instead of ICSI, served as the main insemination method. The amplification rate was consistent with those reported in previous studies. The overall ploidy concordance rates of niPGT-A and PGT-A were 67.7%, while sensitivity and specificity were 57.9% and 83.3%, respectively. The previously reported concordance rates between SCM and TE biopsies are generally heterogeneous, varying from 15.4% to 100% [33]. The concordance rate would be higher if “ploidy concordance” was applied [19]; if each chromosome’s “full concordance” was applied, the concordance would be lower [23]. From this point of view, the niPGT-A can only be applied as a screening tool instead of a precise diagnostic one. Different comparison standards (day 5 TE biopsy or whole blastocysts) produced different results. The sensitivity and specificity of niPGT-A vary among studies [23,24,25,27,28]. Despite its lower sensitivity, niPGT-A may serve as a prioritizing method in advanced maternal age cohorts with high aneuploidy rates and good positive predictive values. Our study revealed adequate specificity and positive predictive values that align with those of a multiple-center study [19] and review article [17].

Euploid embryos were found to have a significantly higher cfDNA amount in the SCM in the advanced maternal age cohort. The nature of increasing aneuploidy with increasing age is well known [47]. As reported by Orvieto et al., 55.5% of euploid blastocysts expel aneuploid debris [48], strongly suggesting that the primary source of cfDNA in culture media is expelled aneuploid blastomeres and/or their fragments. Such self-correction ability must be considered when interpreting the niPGT-A results [35]. In the present study, the more significant cfDNA amount in euploid embryos at the advanced maternal age may be explained by the ability of human embryos to expel the cell fragments or undergo self-correction [48,49,50]. Younger or euploid embryos require less correction, and amplification failure after WGA may be a positive indicator. In a study cohort with an average age of 35.9 years [22], four noisy niPGT-A copy number profiles were identified to be associated with euploid embryos based on the corresponding whole-embryo results. Such “noisy” profiles might be due to the higher gain in whole-genome amplification required for these samples of insufficient DNA (lack of apoptosis) in the SCM. Although the mechanism of secretion is not well-known, the potential sources of cfDNA in the SCM may be derived from cell lysis, apoptosis, or fragmented cellular debris [51]. If the levels of genetic material are related to embryo quality, this may be a potential biomarker for embryo selection [51].

Low cfDNA concentrations, parental contamination, and two peak size profiles were noted in this study. Further, the amplification rate was found to be consistent with that reported in previous studies. According to Xu et al., the total amount of cfDNA in each spent culture is equivalent to that of a fraction of a single cell before amplification [27]. A similar amount of approximately 6.5 pg was reported by Vera-Rodriguez et al. [23]. The mean concentration of amplified cfDNA varied from 15.2 ± 2.7 ng/μL [25] to 58.03 ± 35.87 ng/μL [24] in the SCM after 24–48 h of incubation with blastocysts [33]. This post-amplified mean DNA concentration in this study was 1.97 ng/μL, and the median concentration was 1.5 ng/μL (IQR: 1.0–2.5 ng/μL). The lower concentration may be due to: 1. the higher volume of the culture medium (25–30 μL) used in this study, 2. the incubation time was not as suggested and needed to be longer as per the manufacturer’s protocol. The unstable osmolarity would be a concern if decreasing culture medium volume and extending culture to day 6 was not the IVF routine. Further, a greater amount of amplified DNA would not ensure a better fetal cfDNA quality because parental contamination should be considered. Potential sources of contamination may arise from maternal cumulus cells, sperm, and polar bodies that persist until the blastocyst stage [51]. To minimize the parental contamination, thorough denudation and then ICSI was performed in the majority of previous studies. To prevent cross-contamination, each embryo must be transferred using a new disposable pipette. Moreover, more embryonic cfDNA fractions would be obtained if culturing time was prolonged. Despite having higher concentrations of cfDNA on day 3, high-quality cfDNA was more likely to be successfully amplified in later days’ SCM [26]. To avoid the contamination of cells shed from the operator, wear personal protection equipment including a lab coat, disposable gloves and a face mask when handling the sample(s) and the reagent(s). Stigliani et al. thoroughly characterized cfDNA and measured the linear whole nuclear gDNA amplification product, which had a mean size of 400 bp [52]. We applied a WGA-MALBAC protocol similar to that of Li et al., which resulted in an amplification product of 300–2000 bp [24]. The size distribution of the amplification products in the present study displayed two patterns. A total of 76 out of the 118 revealed a dominant peak at 328.7 bp while the remaining 42 had two dominant peaks: one minor peak at 160 bp and another prominent peak at 321 bp. Among the pattern of two peaks (42/118 = 36%), the minor (smaller size) population may originate from BF DNA [53] while the major (longer size) population may result from other genomic DNA. The size distribution profile of cfDNA revealed the complex composition nature of SCM [54]. Previous studies revealed that artificial shrinkage may protect the blastocyst from membrane-damaging ice crystal formation [40]. Therefore, artificial shrinkage on expanded blastocyst before vitrification was routine in our lab. In recent years, some studies have revealed that blastocoele fluid (BF) was one promising source of embryonic DNA [21,24,29]. However, there were no significantly different DNA concentrations between artificial shrinkage or not (data not shown). Artificial shrinkage may not be required according to this study.

Our study protocol involved less micromanipulation, which may pose a lower risk to the embryos. The problems encountered were not detectable under the non-invasive PGT-A standard ICSI process and extended blastocyst culture. In the present study, ICSI did not improve cfDNA quantity/quality in niPGT-A. The ploidy concordance rate was higher in ICSI samples (ICSI, 11/15 = 73.3%; IVF, 10/16 62.5%; *p* = 0.52), but without statistical difference. Although most insemination methods use ICSI, a recent multi-center study (ICSI was conducted in 90.6% of cycles) reported that the two fertilization methods had similar sensitivity (80.9% vs. 87.9%) and specificity (78.6% vs. 69.9%) [19]. Our study revealed that the interference by cfDNA released by sperms in IVF SCM was also limited after denudation and culture medium replacement on day 3. In addition to the high cost and the need for professional skills, the ICSI procedure could damage cytoplasmic structures (i.e., meiotic spindle) in the oocyte, resulting in sublethal cellular injury [55]. To minimize maternal contamination, enzymatic and mechanical stress during the removal of cumulus cells may cause a high degree of spindle deviation before the ICSI procedure [55]. Briefly, niPGT-A should encompass the IVF method and enrich fetal cfDNA [56,57,58] or the “non-invasive” was always invasive under ICSI.

A time-lapse incubator was introduced for human IVF in the last decade, which is a much later introduction relative to other bioscience fields. Nonetheless, its introduction has led to significant changes in the observation of embryos [12]. Time-lapse morphokinetic parameters can enhance conventional morphological assessments to improve embryo selection and subsequent reproductive outcomes. Automation and artificial intelligence (AI) have recently been introduced to improve this technology [59]. Furthermore, morphokinetic parameters can aid in differentiating between euploid and aneuploid embryos, despite lacking sufficient accuracy to replace the PGT-A. Morphokinetic assessment, chromosomal screening, and AI [60] may help identify euploid embryos with the highest developmental potential. On day 5, embryos with the highest probability of implantation had a CC3 between 9.7 h and 21 h [43]. Our study revealed that CC3 within this range is associated with more chromosomally normal embryos.

The application of niPGT-A in routine clinical settings for diagnostic purposes is not recommended [10]. High rates of false-positive diagnoses of human embryos often lead to non-use and/or disposal of embryos with an entirely normal pregnancy potential [35]. Our results suggest that niPGT-A is a good rule-in assay for identifying normal chromosomal embryos for transfer and might serve as a non-invasive approach prior to invasive TE PGT-A for prioritizing embryos for transfer (Figure 2). With further clinical studies and validations, niPGT-A might provide a safer alternative in embryo screening to improve the clinical outcomes of assisted reproductive technology (ART) [61]. In this study, the embryo transfer strategy was based on a blastocyst morphology evaluation performed in our routine practice. Later, we found that approximately 20% of cycles would change the embryo priority if the physician had access to niPGT-A or time-lapse parameter information in advance (Appendix A).

Our study had some limitations. First, owing to the preciousness of euploid embryos in couples undergoing PGT, the sample size of this study was limited. Second, maternal DNA contamination was a barrier to the accuracy of SCM samples. Further, single-nucleotide polymorphism (SNP) sequencing was not performed.

## 5. Conclusions

Based on this study, performing niPGT-A without altering daily laboratory procedures cannot provide a precise diagnosis. However, niPGT-A can be applied in clinical IVF, enabling the addition of blastocysts with a better prediction of euploidy for transfer. Notably, euploid and aneuploid embryos were found to exhibit different kinetic behaviors. Based on these features, the proposed algorithm may be an additional method to prioritize embryos and increase the probability of non-invasively selecting euploid embryos.

## Figures and Tables

**Figure 1 biomedicines-10-01386-f001:**
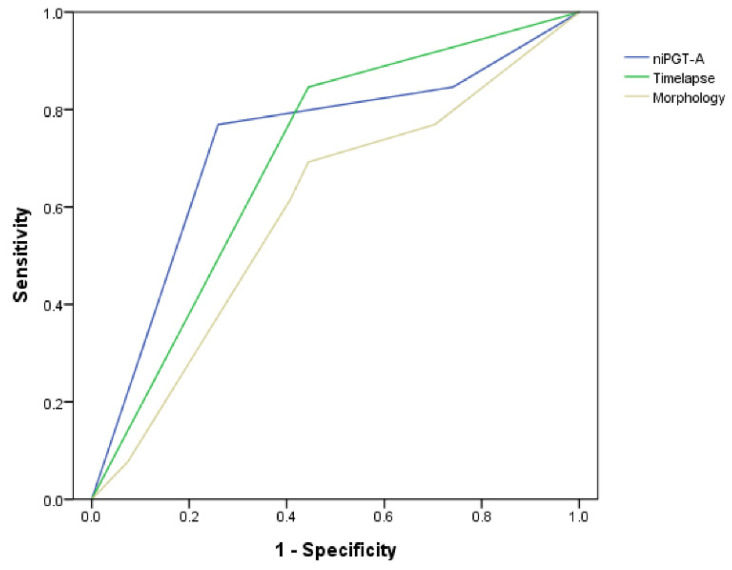
Receiver operating characteristic (ROC) curves of the niPGT-A (AUC = 0.73), time-lapse parameters (AUC = 0.70), and morphology (AUC = 0.59) assessment in predicting euploidy embryos.

**Figure 2 biomedicines-10-01386-f002:**
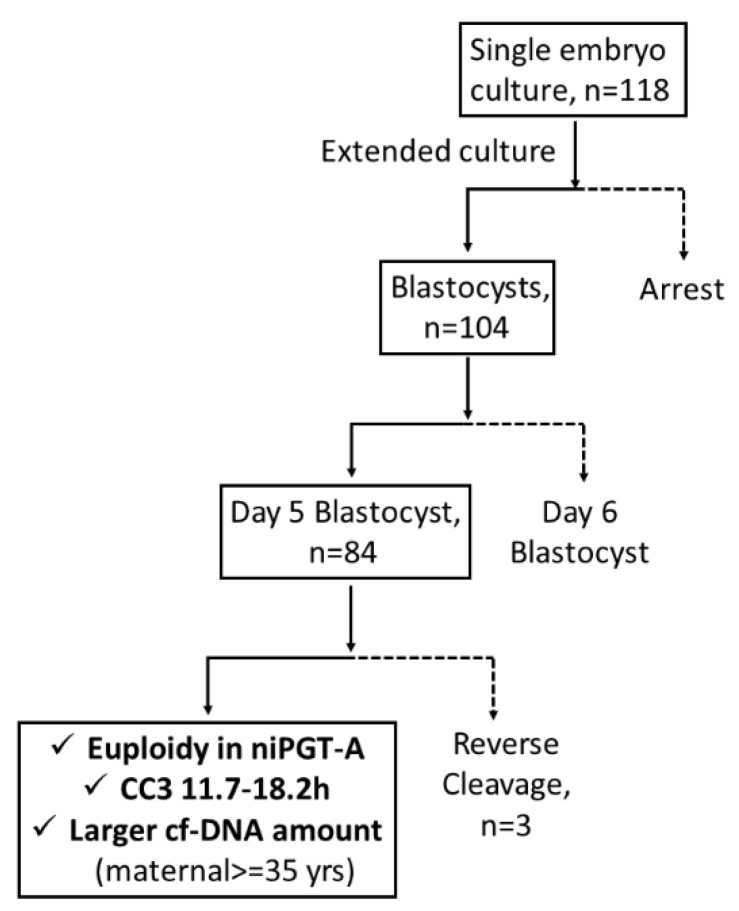
The embryo selection algorithm. Note: Solid line, rule-in; dotted line, rule-out. CC3 = the third cell cycle (T5–T3).

**Table 1 biomedicines-10-01386-t001:** Culture medium (cfDNA) derived from IVF or ICSI.

Variables	IVF, *n* = 75	ICSI, *n* = 43	*p*
Age, year	34.5 (4.9)	35.4 (4.8)	0.35
Qubit, ng/uL	2.1 (1.7)	1.8 (1.3)	0.26
Alive embryo on D5, *n* (%)	69 (92.0%)	38 (88.4%)	0.53
Collected D6, *n* (%)	12 (16.0%)	8 (19.0%)	0.44
QC failed, *n* (%)	9 (12.0%)	3 (7.0%)	0.29
Size peak ≥ 2, *n* (%)	28 (37.3%)	19 (44.2%)	0.56
**Contamination ^1^**			
Global	16 (21.3%)	8 (18.6%)	0.40
Maternal	14 (18.7%)	8 (18.6%)	NS
Paternal	2 (2.7%)	0 (0.0%)	NS

Note: Values are presented as Mean (SD) or *n* (%); NS = non-significant. ^1^ Contamination = Maternal and paternal contamination identified: (1) when discordant sex was observed in SCM compared with TE biopsy or live birth; (2) Aneuploid female in TE biopsy but with euploid DNA in the spent culture media; (3) If the Y copy number falls within the gray zone, then it indicates likely maternal cumulus cell contamination as indicated by the manufacture (the Kit provider) (Yikon Genomics).

**Table 2 biomedicines-10-01386-t002:** Clinical, biological characteristics and NGS results in SCM and TE/whole embryo samples.

Age	Insemination Method	Collection Day	Blastocyst Morphology Grade	Qubit_SCM (ng/uL)	Karyotype_ SCM	Qubit_TE/ Whole Embryo (ng/uL)	Karyotype_ TE/Whole Embryo	Ploidy Concordance ^1^
39	ICSI	6	5BB	4.86	45, X	26.40	47, XY, +1 (×3, mos, ~70%), +16 (×3)	Y
39	ICSI	5	5BB	2.74	46, XX	14.50	46, XX	Y
39	ICSI	5	5AB	2.18	46, XX	21.40	46, XY	Y
40	IVF	6	5AA	0.88	45, X, −X (×1)	25.80	48, XX, +21 (×3), +22 (×3)	Y
40	IVF	6	5AB	1.50	46, XX	22.20	45, XY, −15 (×1)	N
40	IVF	6	5BB	1.42	46, XX	22.40	45, X, −X (×1), +2 (×3), −14 (×1)	N
38	IVF	5	5BB	2.14	45, XY, −13 (×1)	8.24	46, XY, +10 (×3, mos, ~50%)	N
38	IVF	6	5AA	4.32	46, XX	20.40	46, XX	Y
38	IVF	5	5AB	1.67	43, X, −X (×1), −10 (×1), −20 (×1)	23.00	47, XX, +21 (×3)	Y
38	IVF	5	5BB	1.99	47, XX, +14 (×3)	24.40	46, XX	N
39	ICSI	6	5CC	1.96	49, X, +12 (×3), +17 (×4), +22 (×3)	high	48XY, +3 (×3), +22 (×3)	Y
39	ICSI	5	5AA	1.19	46, XX	high	47XX, +3 (×3)	N
39	ICSI	6	5AB	0.86	46, XX	high	XY, Mosaic partial duplication of chromosome 14, 50%	N
32	ICSI	5	5AB	0.45	46, XX	high	46, XX	Y
32	ICSI	6	5AB	4.10	46, XX	high	46, XX	Y
32	ICSI	6	5BC	3.86	46, XX	high	XX, Multiple chromosomal abnormalities	N
41	ICSI	6	5CC	1.92	46, XX, −2 (×1), −5 (×1), +7 (×3), +14 (×3)	high	XX, Monosomy 2 & 5; Trisomy 7; Mosaic trisomy 14	Y
41	ICSI	6	5BB	0.82	46, XX	high	47, XY, +20 (×3)	N
40	IVF	5	5AA	0.48	QC fail	14.80	45, XY, −22 (×1)	N/A
40	IVF	5	5AB	0.45	QC fail	13.00	47, XY, +22 (×3)	N/A
40	IVF	5	5AB	0.69	46, XY	13.60	46, XY, −16 (×1), +17 (×3)	N
40	IVF	5	5BB	0.38	QC fail	15.40	46, XY, +16 (×3), −22 (×1)	N/A
40	IVF	5	5BB	0.17	QC fail	15.10	45, XX, −10 (×1)	N/A
40	IVF	5	5AB	1.96	46, XX	11.30	46, XX	Y
40	IVF	5	5AB	1.81	46, XX	11.00	46, XY	Y
40	IVF	5	5AA	3.14	46, XX	9.58	46, XX	Y
40	IVF	5	5AB	1.03	47, XX, +16 (×3)	17.40	45, XX, −16 (×1)	Y
40	IVF	5	5BA	1.61	46, XX	8.74	46, XX	Y
40	IVF	5	5BB	0.59	QC fail	10.80	48, XX, +6 (×3), +15 (×3)	N/A
39	ICSI	3	Arrest	1.48	44, XY, −2 (×1), −18 (×1)	22.40	51, XX, +10 (×3), +18 (×3), +20 (×3), +22 (×4)	Y
39	ICSI	3	Arrest	3.70	48, XX, +1 (×3), +18 (×3)	26.60	47, XX, +18 (×3)	Y
39	ICSI	3	Arrest	1.53	45, X	26.00	47, XXY, +X (×2)	Y
40	IVF	3	Arrest	1.05	47, XXY, +X (×2)	33.40	48, XXY, +X (×2), +13 (×3)	Y
40	IVF	3	Arrest	2.32	46, XX	25.80	46, XY	Y
40	IVF	3	Arrest	1.65	46, XX, −8 (×1), +13 (×3)	21.80	47, XXY, +X (×2), +11 (×3), +15 (×3), −21 (×1)	Y
40	IVF	3	Arrest	0.69	QC fail	29.00	46, XY	N/A
40	IVF	3	Arrest	0.12	QC fail	6.44	46, XX	N/A
40	IVF	3	Arrest	1.57	QC fail	26.60	47, XXY, +X (×2)	N/A
40	IVF	3	Arrest	0.55	QC fail	24.00	50, X, −Y(×0), Multiple chromosomal abnormalities	N/A
40	IVF	3	Arrest	3.48	46, XX	26.40	44, XX, −14 (×1), −19 (×1)	N

SCM = spent culture medium. ^1^ Ploidy concordance, when the result from the trophectoderm biopsy and the SCM were euploid–euploid or aneuploid–aneuploid from the two sample types. Y = concordant. N = disconcordant. N/A = non-applicable.

**Table 3 biomedicines-10-01386-t003:** Time-lapse parameters incorporated in assessing embryos.

No.	Day	Blastocyst Grading	TE PGT-A	niPGT-A (SCM)	CC2 (T3–T2)	CC3 (T5–T3)	T5–T2	S2 (T4–T3)	Reverse Cleavage
1	D5	5AB	Euploidy	Euploidy					
2	D6	5AB	Euploidy	Euploidy					
3	D6	5AA	Euploidy	Euploidy					
4	D5	5AB	Euploidy	Euploidy					
5	D5	5AA	Euploidy	Euploidy					
6	D5	5BB	Euploidy	Euploidy					
7	D5	5AB	Euploidy	Euploidy					
8	D5	5AB	Euploidy	Euploidy					
9	D5	5BA	Euploidy	Euploidy					
10	D5	5BB	Euploidy	Aneuploidy					
11	D5	5BB	Mosaicism	Aneuploidy					
12	D5	5AA	Aneuploidy	Euploidy					
13	D6	5BC	Aneuploidy	Euploidy					
14	D6	5BB	Aneuploidy	Euploidy					
15	D5	5AB	Aneuploidy	Euploidy					
16	D5	5AB	Aneuploidy	Aneuploidy					
17	D6	5BB	Mosaicism	Aneuploidy					
18	D6	5AA	Aneuploidy	Aneuploidy					
19	D6	5AB	Mosaicism	Euploidy					
20	D6	5CC	Mosaicism	Aneuploidy					
21	D5	5AB	Aneuploidy	Aneuploidy					
22	D6	5BB	Aneuploidy	Euploidy					
23	D5	5AB	Aneuploidy	Aneuploidy					
24	D5	5BB	Aneuploidy	QC fail					
25	D5	5AA	Aneuploidy	QC fail					
26	D5	5AB	Aneuploidy	QC fail					
27	D5	5BB	Aneuploidy	QC fail					
28	D5	5BB	Aneuploidy	QC fail					
29	D6	5CC	Aneuploidy	Aneuploidy					
30		Arrest	Euploidy	QC fail					
31		Arrest	Euploidy	QC fail					
32		Arrest	Euploidy	Euploidy					
33		Arrest	Aneuploidy	Aneuploidy					
34		Arrest	Aneuploidy	Aneuploidy					
35		Arrest	Aneuploidy	Aneuploidy					
36		Arrest	Aneuploidy	Aneuploidy					
37		Arrest	Aneuploidy	Aneuploidy					
38		Arrest	Aneuploidy	QC fail					
39		Arrest	Aneuploidy	Euploidy					
40		Arrest	Aneuploidy	QC fail					

The black-filled grid represents abnormal intervals and/or positive reverse cleavage. CC2, calculated as T3–T2, the second round of cleavage; CC3, calculated as T5–T3, the third round of cleavage; S2, the synchrony in the division from two to four cells.

## Data Availability

The datasets used and/or analyzed during the current study are available from the corresponding author on reasonable request.

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
