# Peer review of "Validation of Non-Invasive Preimplantation Genetic Screening Using a Routine IVF Laboratory Workflow"

_biomedicines, 2022, doi:10.3390/biomedicines10061386_

Round 1

Reviewer 1 Report

This study assessed ni-PGT-A of SCM to determine if the authors could detect concordance with traditional TE biopsy with PGT-A. This study did not control for IVF vs ICSI and did not modify the media drop size for ni-PGT-A. The study only reported the ploidy concordance rate rather than reporting the full concordance. In fact, none of the aneuploid samples analyzed yielded full concordance which is very troubling and is suggestive of an issue with the ni-PGT-A analysis used here. Additionally, no mosaics were reported with the TE/PGT-A results which is a little odd considering how prevalent mosacism is with the TE. The methods are also unclear in sentence 1 of the “spent culture medium collection” section. This section of methods also described collection of blast fluid mixed with culture media, but these results were not provided in the paper. Minor issue: The table is hard to read with tables spanning multiple pages lacking headings.

Author Response

Point 1: This study assessed ni-PGT-A of SCM to determine if the authors could detect concordance with traditional TE biopsy with PGT-A. This study did not control for IVF vs ICSI and did not modify the media drop size for ni-PGT-A. The study only reported the ploidy concordance rate rather than reporting the full concordance. In fact, none of the aneuploid samples analyzed yielded full concordance which is very troubling and is suggestive of an issue with the ni-PGT-A analysis used here.

Response 1: Thank you for your comment and we agree with your point. Overall, the full concordance rates were lower compared with the ploidy concordance rates in the previous studies. According to the previous literature, the ploidy concordance rate (or diagnostic concordance rate) of SBM and the TE biopsy, ranged from 33.3% [1] to 89.1% [2]. The full concordance rates ranged from 17.6% [1] to 67.7% [3]. From this point of view, the niPGT-A can only be applied as a screening tool instead of a precise diagnostic one.

The full concordance rate in the present study was 8/31=25.8%. If niPGT-A was viewed as a prioritizing tool, then we only need to rule in euploid embryos instead of knowing every detail of abnormal chromosomes.

We added the full concordance rate to the manuscript (Results 3.2.) and in Discussion (Line 237, 298). Really appreciated your suggestions.

Point 2: Additionally, no mosaics were reported with the TE/PGT-A results which is a little odd considering how prevalent mosacism is with the TE.

Response 2: Indeed, the Mosaicism% in TE biopsy with the introduction of NGS was not uncommon. The incidence of mosaicism in blastocysts ranged from 1 to 40% [4]. We re-analysis the NGS report in ChromoGo software (as mentioned in the methods section, 2.5.). The reported results revealed that 4/40= 10% of TE/PGT-A was mosaicism. We edited Table 2. in the “karyotype_TE” column and “TE PGT-A” column in Table 3. The mosaicism rate was also added in the Results section (3.2.)(Line 226).

Point 3: The methods are also unclear in sentence 1 of the “spent culture medium collection” section. This section of methods also described collection of blast fluid mixed with culture media, but these results were not provided in the paper.

Response 3: The timing of SCM collection was mainly on day 5, unless the embryos were delayed for blastulation. We routinely replaced the culture medium on day 3 with G2. Therefore, the incubation times in this study was 2 days for day 5 blastocysts, and 3 days for day 6 blastocysts. We didn’t conduct blastocentesis [1] to collect blastocoel fluid (BF). Instead, the artificial shrinkage with laser over the cellular junction of the TE was performed on expanded blastocyst before vitrification [2] and this would  release the blastocoel fluid into the culture medium. We edited the sentences in section of “spent culture medium collection (2.3.) (Line 111-114, 124)”.

Point 4: Minor issue: The table is hard to read with tables spanning multiple pages lacking headings.

Response 4: Thanks for your reminder. We have edited and adjust the mega Table, making it easier for readers.

References

  1. Shi, W.; Zhao, Z.; Xue, X.; Li, Q.; Yao, Y.; Wang, D.; Wang, J.; Lu, S.; Shi, J. Ploidy Testing of Blastocoel Fluid for Screening May Be Technically Challenging and More Invasive Than That of Spent Cell Culture Media. Front Physiol 2022, 13, 794210, doi:10.3389/fphys.2022.794210.
  2. Vanderzwalmen, P.; Bertin, G.; Debauche, C.; Standaert, V.; van Roosendaal, E.; Vandervorst, M.; Bollen, N.; Zech, H.; Mukaida, T.; Takahashi, K., et al. Births after vitrification at morula and blastocyst stages: effect of artificial reduction of the blastocoelic cavity before vitrification. Hum Reprod 2002, 17, 744-751, doi:10.1093/humrep/17.3.744.

Reviewer 2 Report

Shortening of the duration of assisted reproduction (ART) and reducing the incidence of multiple pregnancies both rely on the identification of those embryos that have the highest potential for proper implantation and healthy pregnancy. The authors of this in principle interesting study have embarked on comparing four different methods of embryo assessment: 1. embryonic morphology 2. timelapse assessment 3. PGT-A of trophectoderm biopsy material, and 4. non-invasive PGT (niPGT-A) of spent culture medium (SCM). This presumably prospective cohort study was initially performed on 118 SCM obtained from 38 infertile couples undergoing ART, but in only 40 SCM information of the genetic composition of the embryos was available. Ultimately, full data coverage was available in 40 cases only.

Although the authors come up with a valuable study question, the design of the study and the presentation of the results are incoherent and render an in depth interpretation of the data rather difficult.

What was the primary design of the study? Is this a prospective cohort study? If so, was the design of the study presented to and approved by the local ethics committee? I was unable to find any reference to an approval by a local ethical board.

Section 2.7 Embryo transfer: what is meant with “the results were initially blinded to the primary care doctor”? Why “initially”? On which basis were embryos selected for embryo transfer and how many embryos were transferred per treatment cycle? If the treating physicians were blinded, this would mean that this was a prospective study based on a predefined study design. If so, this must be declared at the start of the Material and Methods section (2.1). 

The study was performed from March 2019 to July 2020. According to the statement in the manuscript (lines 99 to 101) no couples were excluded. How can during this prolonged time interval only 32 couples have been included, if no couples were excluded? Please explain.

Abstract: the last sentence of the abstract does not fit to the statements in the conclusion at the end of the discussion. At this stage of data analysis, niPGT-A cannot be used to prioritize ranking of embryos, as correctly stated in the abstract. Furthermore, the authors promised evaluation of diagnostic accuracy (with sensitivity and specificity analyses of the obtained results, as stated in the statistical analysis), but failed to present any data on that. Ideally, the authors should compare the four different embryo assessment methods based on receiver operating characteristics (ROC). A ROC-analysis would have been a logical consequence of the statistical analyses, outlined in section 2.8, lines 168 to 171. This could be executed by limiting the analysis of the data to the 40 SCM for which all information is available (Table 3 and 4).

A number of items used throughout the manuscript have not been explained towards the interested reader. I understand that the measure qubit results from the qubit high sensitivity dsDNA kit, but the authors should explain what the measured item qubit really means. Furthermore, QC: please explain what is meant with QC?

Figure 1 represents a flow chart of the study design. Instead of using the term “single culture embryos” the term “single embryo culture” seems preferable. Figure 1 also should be implemented with the real numbers in each section, in order to understand how the embryo or material selection process went on.

The in vitro culture was performed in a non-sequential culture medium, as described in section 2.2 of the manuscript. This results in the embryos growing from day 3 onward in a distinct culture medium (G2, Vitrolife), that was obviously used for SCM. As a result, the embryos had only two or three days to impregnate the culture medium for subsequent DNA analysis. The authors should elaborate on that timely limitation, because this is an important technical requirement for niPGT-A. It must also be considered in the light of comparing the results of niPGT-A after conventional IVF and ICSI.

Another important and crucial requirement is the exclusion of SCM contamination with operator DNA during embryo manipulation. How did the authors exclude his potential source of foreign DNA contamination?

Finally, the authors breached the expanding blastocyst embryos prior to vitrification using laser. This invasive procedure was justified by the authors to maximize DNA content in SCM for niPGT-A. This invasive intervention stands in contrast to the supposed non-invasive nature of niPGT-A. The authors are required to reference their justification of such an invasive step and to discuss this “requirement” in the discussion.

Minor issues:

Line 118: I guess that the authors meant “elective single blastocyst transfer”.

Line 119: I guess: to laser instead of to lase.

Line 126: I guess that the blank control consisted of embryo culture medium without contact to the embryo. The meaning of this sentence is ambiguous and should be formulated better.

Line 160: cc2 <=11.9h and cc3 11.7-18.2h: there is a timely overlap here. Please correct or explain.

Lines 368 and following: please remove that sentence, which is far from being neutral.

Reviewer 3 Report

In the manuscript, the author reported clinical results of niPGT-A for embryo quality assessment. Experimental results of SCM niPGT-A is reported and compared with TE PGT, and the combination of time-lapse morphokinetics with niPGT-A is discussed. 

Detailed comments:

1. DNA concentration in TE biopsy results is very low. The author should discuss the reason of this phenomenon. 

2. It is necessary to discuss the difference between the IVF and ICSI embryo niPGT-A results.

3. The author should discuss the influence of DNA contamination to the niPGT process and the corresponding sample treatment process to minimize contamination. 

Author Response

In the manuscript, the author reported clinical results of niPGT-A for embryo quality assessment. Experimental results of SCM niPGT-A is reported and compared with TE PGT, and the combination of time-lapse morphokinetics with niPGT-A is discussed. Detailed comments:

Point 1: DNA concentration in TE biopsy results is very low. The author should discuss the reason of this phenomenon.

Response 1: The low cfDNA concentration may be due to: 1. The volume of the culture medium was not reduced, 2. The incubation time was not as suggested to be longer as the manufacturer’s protocol. The unstable osmolarity would be a concern if decreasing culture medium volume and extending culture to day 6 was not the IVF routine.

We edited the content in Discussion section (Line 331-335). Thanks for your suggestions.

Point 2: It is necessary to discuss the difference between the IVF and ICSI embryo niPGT-A results.

Response 2: There were 63.6% (n=75) SCM derived from IVF and 36% from ICSI. The maternal age was comparable between the two groups (IVF, 34.5 y/o; ICSI, 35.4 y/o; p=0.35). The DNA amount after WGA revealed not significantly different between IVF (2.1 ng/uL) and ICSI (1.8 ng/uL). Failed QC rate was not significantly higher in IVF group. Global parental DNA contamination was observed with 21.3% in IVF and 18.6% in ICSI (p=0.397). Maternal as well as paternal contamination were noticed in IVF groups whereas no paternal contamination was noted in ICSI ones. The ICSI provides no better cfDNA quantity/quality in niPGT-A under routine IVF workflow. The ploidy concordance rate was higher in ICSI samples (ICSI, 11/15=73.3%; IVF, 10/16 62.5%; p=0.52), but without stastiscally difference.

In the multicenter prospective study (ICSI was conducted in 90.6% of cycles) [1], the two fertilization techniques provided similar sensitivity (80.9% vs 87.9%) and specificity (78.6% vs 69.9%). Furthermotre, in Shi et al. study [2], the overall ploidy concordance in IVF and ICSI was comparable (84.4 versus 84.5%, p = 0.983) in SCM of thawed blastocysts. Our study revealed that the interference by cfDNA released by sperms in IVF SCM was also limited after denudation and culture medium replacement on day 3.

We have edited the previous paragraph to Results (Line 199-203) and Discussion section (Line 357-363).

Point 3: The author should discuss the influence of DNA contamination to the niPGT process and the corresponding sample treatment process to minimize contamination.

Response 3: Potential sources of contamination may arise from maternal cumulus cells, sperm, and polar bodies that persist until the blastocyst stage. To minimize the parental contamination, thorough denudation and then ICSI was performed in the majority of previous studies. To prevent cross-contamination, each embryo must be transferred using a new disposable pipette. Besides, more fetal cfDNA fraction would be obtained if prolonged culturing time. Despite having higher concentrations of cfDNA on day 3, high-quality cfDNA was more likely to be successfully amplified in later days’ SCM [3]. In the future, enrichment of fetal cfDNA in niPGT-A is a crucial direction to circumventing contaminations [4,5].

We have added the previous paragraph to Discussion section (Line 338-343).

References

  1. Rubio, C.; Navarro-Sánchez, L.; García-Pascual, C.M.; Ocali, O.; Cimadomo, D.; Venier, W.; Barroso, G.; Kopcow, L.; Bahçeci, M.; Kulmann, M.I.R., et al. Multicenter prospective study of concordance between embryonic cell-free DNA and trophectoderm biopsies from 1301 human blastocysts. Am J Obstet Gynecol 2020, 223, 751.e751-751.e713, doi:10.1016/j.ajog.2020.04.035.
  2. Shi, W.; Zhao, Z.; Xue, X.; Li, Q.; Yao, Y.; Wang, D.; Wang, J.; Lu, S.; Shi, J. Ploidy Testing of Blastocoel Fluid for Screening May Be Technically Challenging and More Invasive Than That of Spent Cell Culture Media. Front Physiol 2022, 13, 794210, doi:10.3389/fphys.2022.794210.
  3. Ho, J.R.; Arrach, N.; Rhodes-Long, K.; Ahmady, A.; Ingles, S.; Chung, K.; Bendikson, K.A.; Paulson, R.J.; McGinnis, L.K. Pushing the limits of detection: investigation of cell-free DNA for aneuploidy screening in embryos. Fertil Steril 2018, 110, 467-475 e462, doi:10.1016/j.fertnstert.2018.03.036.
  4. Liang, B.; Li, H.; He, Q.; Li, H.; Kong, L.; Xuan, L.; Xia, Y.; Shen, J.; Mao, Y.; Li, Y., et al. Enrichment of the fetal fraction in non-invasive prenatal screening reduces maternal background interference. Sci Rep 2018, 8, 17675, doi:10.1038/s41598-018-35738-0.
  5. Qiao, L.; Yu, B.; Liang, Y.; Zhang, C.; Wu, X.; Xue, Y.; Shen, C.; He, Q.; Lu, J.; Xiang, J., et al. Sequencing shorter cfDNA fragments improves the fetal DNA fraction in noninvasive prenatal testing. Am J Obstet Gynecol 2019, 221, 345.e341-345.e311, doi:10.1016/j.ajog.2019.05.023.

Round 2

Reviewer 2 Report

The authors have improved their manuscript according to the lines that were given in my first review. The addition of the ROC curves have improved the article. Although the manuscript is still not perfect, I am inclined to accept it in its present form.